# Liver Function Tests in COVID-19: Assessment of the Actual Prognostic Value

**DOI:** 10.3390/jcm11154490

**Published:** 2022-08-01

**Authors:** Urszula Tokarczyk, Krzysztof Kaliszewski, Anna Kopszak, Łukasz Nowak, Karolina Sutkowska-Stępień, Maciej Sroczyński, Monika Sępek, Agata Dudek, Dorota Diakowska, Małgorzata Trocha, Damian Gajecki, Jakub Gawryś, Tomasz Matys, Justyna Maciejiczek, Valeriia Kozub, Roman Szalast, Marcin Madziarski, Anna Zubkiewicz-Zarębska, Krzysztof Letachowicz, Katarzyna Kiliś-Pstrusińska, Agnieszka Matera-Witkiewicz, Michał Pomorski, Marcin Protasiewicz, Janusz Sokołowski, Barbara Adamik, Krzysztof Kujawa, Adrian Doroszko, Katarzyna Madziarska, Ewa Anita Jankowska

**Affiliations:** 1Clinical Department of General, Minimally Invasive and Endocrine Surgery, Wroclaw Medical University, Borowska Street 213, 50-556 Wroclaw, Poland; karolina.sutkowska@onet.pl (K.S.-S.); maciej.sroczynski@o2.pl (M.S.); moniasep@wp.pl (M.S.); agatadudek7@gmail.com (A.D.); 2Statistical Analysis Center, Wroclaw Medical University, Marcinkowski Street 2-6, 50-368 Wroclaw, Poland; anna.kopszak@umw.edu.pl (A.K.); krzysztof.kujawa@umw.edu.pl (K.K.); 3Clinical Department of Urology and Urological Oncology, Wroclaw Medical University, Borowska Street 213, 50-556 Wroclaw, Poland; lllukasz.nowak@gmail.com; 4Department of Basic Science, Faculty of Health Science, Wroclaw Medical University, Bartel Street 5, 51-618 Wroclaw, Poland; dorota.diakowska@umw.edu.pl; 5Department of Pharmacology, Wroclaw Medical University, Mikulicz-Radecki Street 2, 50-345 Wroclaw, Poland; malgorzata.trocha@umw.edu.pl; 6Clinical Department of Internal and Occupational Diseases, Hypertension and Clinical Oncology, Wroclaw Medical University, Borowska 213, 50-556 Wroclaw, Poland; damian.gajecki@umw.edu.pl (D.G.); jakub.gawrys@umw.edu.pl (J.G.); tomasz.matys@umw.edu.pl (T.M.); adrian.doroszko@umw.edu.pl (A.D.); 7Clinical Department of Internal Medicine, Pneumology and Allergology, Wroclaw Medical University, M. Skłodowskiej-Curie Street 66, 50-369 Wroclaw, Poland; justynamaciejiczej@gmail.com (J.M.); vkozub@usk.wroc.pl (V.K.); romanszalast@gmail.com (R.S.); 8Clinical Department of Rheumatology and Internal Medicine, Wroclaw Medical University, Borowska Street 213, 50-556 Wroclaw, Poland; madziarski.marcin@gmail.com; 9Clinical Department of Gastroenterology and Hepatology, Wroclaw Medical University, Borowska Street 213, 50-556 Wroclaw, Poland; anna.zubkiewicz-zarebska@umw.edu.pl; 10Clinical Department of Nephrology and Transplantation Medicine, Wroclaw Medical University, Borowska Street 213, 50-556 Wroclaw, Poland; krzysztof.letachowicz@umw.edu.pl (K.L.); katarzyna.madziarska@umw.edu.pl (K.M.); 11Clinical Department of Pediatric Nephrology, Wroclaw Medical University, Borowska Street 213, 50-556 Wroclaw, Poland; katarzyna.kilis-pstrusinska@umw.edu.pl; 12Screening Laboratory of Biological Activity Assays and Collection of Biological Material, Faculty of Pharmacy, Wroclaw Medical University, Borowska Street 211A, 50-556 Wroclaw, Poland; agnieszka.matera-witkiewicz@umw.edu.pl; 13Clinical Department of Gynecology and Obstetrics, Wroclaw Medical University, Borowska Street 213, 50-556 Wroclaw, Poland; michal.pomorski@umw.edu.pl; 14Clinical Department and Clinic of Cardiology, Wroclaw Medical University, Borowska Street 213, 50-556 Wroclaw, Poland; marcin.protasiewicz@umw.edu.pl; 15Department of Emergency Medicine, Wroclaw Medical University, Borowska Street 213, 50-556 Wroclaw, Poland; janusz.sokolowski@umw.edu.pl; 16Clinical Department of Anaesthesiology and Intensive Therapy, Wroclaw Medical University, Borowska Street 213, 50-556 Wroclaw, Poland; barbara.adamik@umw.edu.pl; 17University Hospital in Wroclaw, Institute of Heart Diseases, Wroclaw Medical University, Borowska Street 213, 50-556 Wroclaw, Poland; ewa.jankowska@umw.edu.pl

**Keywords:** COVID-19, SARS-CoV-2, severity of COVID-19, hospitalized patients, risk factors, liver

## Abstract

Deviations in laboratory tests assessing liver function in patients with COVID-19 are frequently observed. Their importance and pathogenesis are still debated. In our retrospective study, we analyzed liver-related parameters: aspartate aminotransferase (AST), alanine aminotransferase (ALT), alkaline phosphatase (ALP), gamma-glutamyltransferase (GGT), total bilirubin (TBIL), albumin, comorbidities and other selected potential risk factors in patients admitted with SARS-CoV-2 infection to assess their prognostic value for intensive care unit admission, mechanical ventilation necessity and mortality. We compared the prognostic effectiveness of these parameters separately and in pairs to the neutrophil-to-lymphocyte ratio (NLR) as an independent risk factor of in-hospital mortality, using the Akaike Information Criterion (AIC). Data were collected from 2109 included patients. We created models using a sample with complete laboratory tests *n* = 401 and then applied them to the whole studied group excluding patients with missing singular variables. We estimated that albumin may be a better predictor of the COVID-19-severity course compared to NLR, irrespective of comorbidities (*p* < 0.001). Additionally, we determined that hypoalbuminemia in combination with AST (OR 1.003, *p* = 0.008) or TBIL (OR 1.657, *p* = 0.001) creates excellent prediction models for in-hospital mortality. In conclusion, the early evaluation of albumin levels and liver-related parameters may be indispensable tools for the early assessment of the clinical course of patients with COVID-19.

## 1. Introduction

Coronavirus disease 2019 (COVID-19) caused by Severe Acute Respiratory Syndrome Coronavirus 2 (SARS-CoV-2) was first reported to the World Health Organization on 30 December 2019 in Wuhan. [1,2,3]. The symptoms of COVID-19 are predominantly related to the pulmonary tract and are manifested by dry cough, fever, fatigue and headache [4]. However, the virus may also lead to a systemic and multi-organ disease causing extra-pulmonary manifestations involving the cardiovascular, hematological, renal, gastrointestinal and hepatobiliary, neurological, ophthalmological and dermatological systems [5]. According to the meta-analysis, the most common gastrointestinal manifestations include anorexia, diarrhea and nausea. Additionally, vomiting, abdominal pain and abdominal distension can be observed [6].

The liver’s involvement in patients with COVID-19 is still the subject of dispute, especially due to the fact that 76% of patients present liver biochemistry abnormalities, usually mild to moderate [7,8]. A multicenter retrospective study by Wuhan revealed that acute liver injury (ALI) occurs later in the course of COVID-19 (day 17 IQR, 13–23) and follows the development of ARDS [9]. The latest research shows that ALI is more common than initially thought and may occur in up to 22.8% of patients [10]. The potential mechanism of hepatocyte damage remains unclear. Direct damage can be mediated by angiotensin converting enzyme 2 (ACE2) receptors and cellular serine protease called TMPRSS2, which is needed to prime the Spike protein for cell entrance [11]. These receptors are expressed in only 3% of hepatocytes, whereas their presence in cholangiocytes reaches 60% of cells [12,13,14]. However, L-SIGN, which is a liver-specific membrane receptor binding to ACE2, may be excessively expressed in infected sinusoid cells [15]. Thus, it may constitute a bridge for SARS-CoV-2 to infect hepatocytes. Other considered receptors include CD147, which can be overexpressed in an inflammatory process. The affinity between CD147 and the SARS-CoV-2 spike protein was shown in in vitro studies [16]. Sun et al. suggest probable patomechanisms as immune-mediated damage, hypoxic damage and drug-induced liver injury [17]. Additionally, studies focused on the impact of preexisting non-alcoholic fatty liver disease (NAFLD) on the severe course of COVID-19 suggest that obesity associated with NAFLD can contribute to the polarization of macrophages into M1 proinflammatory macrophages, which may exacerbate SARS-CoV-2 infection [18]. Thus, it seems that the liver impairment caused by COVID-19 is likely of multifactorial origin.

The substantial majority of studies focused on COVID-19-associated liver abnormalities evaluate peak levels of liver enzymes, such as aspartate aminotransferase (AST), alanine aminotransferase (ALT), alkaline phosphatase (ALP), gamma-glutamyl transferase (GGT), total bilirubin (TBIL) and albumin [9,19,20]. They showed a predominance of parenchymal liver injury based on the prevalence of elevated AST and ALT [21]. Studies considering liver biochemistry at baseline seem to confirm this tendency [22,23]. Significant elevations in ALP are uncommonly recorded despite strong cholangiocyte ACE2 expression, with mildly raised GGT levels seen in up to 50% of patients [7,22,23,24]. Nevertheless, the clinical significance of these observations remains inconclusive. There are still some lacking answers, especially regarding the impact of previously taken medications.

The clinical course and outcome of patients with COVID-19 and liver abnormalities require further investigation, given the alterations in liver function tests and liver impairment in pathological findings in patients with COVID-19. Furthermore, the association between liver enzymes and worse outcomes, including COVID-19-related in-hospital fatalities, should also be analyzed.

In our retrospective study, we aimed to develop a predictive model for COVID-19 patients based on baseline data, including liver abnormalities especially associated with potential biliary tract damage, hoping to find a new objective path to forecast the progression of COVID-19 to severe and lethal forms while excluding the influence of comorbidities.

## 2. Materials and Methods

### 2.1. Study Population and Data Collection

All procedures were performed in accordance with the ethical standards of Wroclaw Medical University (Poland) and with the 1964 Helsinki Declaration and its later amendments. The study protocol was approved by the Commission of Bioethics at Wroclaw Medical University (No: KB-444/2021).

This study enrolled 2184 adult patients admitted to the University Hospital in Wroclaw (USW) with positive SARS-CoV-2 real-time PCR (RT-PCR) from March 2020 to November 2021. Generic data included the following characteristics: gender, age, prior history of hypertension, diabetes, asthma, chronic obstructive pulmonary disease (COPD), dementia, stroke/transient ischemic attack (TIA), chronic kidney disease, myocardial infraction, heart failure, chronic liver disease, solid malignant disease, leukemia, lymphoma and active acquired immunodeficiency syndrome (AIDS). Further information about smoking status and length of hospital stay was collected. Patients with accompanying chronic liver disease, such as chronic hepatitis, cirrhosis with or without portal hypertension, and fatty liver disease (*n* = 74), were excluded from the analysis. Admission values of laboratory tests including liver enzymes such as AST, ALT, ALP, GGT, TBIL and albumin were examined as a main subject of our survey. Further data collection included a complete blood count with differential used to calculate neutrophil-to-lymphocyte ratio (NLR), which is an independent risk factor of in-hospital mortality [25]. Endpoints for COVID-19 severity were defined as: (1) admission to the ICU, (2) intubation and (3) death. 

### 2.2. Statistical Analysis

Categorical variables are presented as numbers and percentages. Means and standard deviations were used to present the central tendency of the continuous variables. The normal distribution of these variables was verified using the Shapiro–Wilk test. The chi-square test was used to determine the association between the two categorical variables. Reviewing the percentage of patients whose parameters related to liver function were abnormal, we considered only the group without liver diseases in a previous history (*n* = 2109). To analyze the association between liver function tests and the severity of disease, logistic regression was used adjusting for comorbidities: hypertension, diabetes, asthma, COPD, dementia, stroke/TIA in patient history, chronic kidney disease, myocardial infarction in patient history, heart failure, leukemia, lymphoma and solid malignant disease in order of additional assessment of their influence on COVID-19 severity in our population and a potential ability to interfere with the objective assessment of liver-related parameters as predictors. Additionally, adjustment was performed for length of hospital stay and smoking history, which is considered an independent risk factor for worse outcomes in COVID-19 disease [26]. In the above-mentioned logistic regression, we assessed only patients who had complete laboratory tests, which is the reason for the sample size variation (*n* = 401). In further analyses, we created models composed of pairs formed by laboratory parameters (AST, ALT, ALP, GGT, TBIL, albumin) in various configurations using the same adjustments as in the previous analysis to find the best predictors that could be objective laboratory indicators of the severe course of COVID-19, regardless of comorbidities and other factors mentioned above. The accuracy of multivariate models evaluating single parameters related to the liver and their double combinations was assessed using the Nagelkerke coefficient. The best models were selected using the Akaike Information Criterion (AIC) and then applied to the whole sample (*n* = 2109), excluding patients with missing data. Receiver operating characteristic (ROC) curves were calculated to present a comparison between the models.

All statistical analyses were performed using the Statistica 13.3 package and R software, version 4.1.1 (2021-08-10, R Foundation for Statistical Computing). Statistical significance was determined at *p* < 0.05.

## 3. Results

### 3.1. Clinical Characteristics

The baseline clinical characteristics of the COVID-19 patients from our cohort are summarized in Table 1. The median age was 64 years and 50.4% were female. In our sample, men statistically significantly more frequently met fatal outcomes, were admitted to ICU and required invasive respiratory support. The most prevalent underlying medical conditions were arterial hypertension (46.8%), diabetes mellitus of any kind (23.6%), heart failure (11.7%) and chronic kidney disease (10.6%). Hypertension and diabetes were significantly more frequent in the groups for all endpoints. Furthermore, myocardial infraction in the patient history co-occurred with mechanical ventilation requirements and fatal outcomes. In the group of deceased patients, the prevalence of dementia, stroke/TIA in the past, chronic kidney disease, heart failure and solid malignant disease was also statistically relevant.

### 3.2. Clinical Course and Outcome

Out of the 2184 patients with SARS-CoV-2 infection that were included in our cohort, 214 (9.8%) required treatment in the ICU and 215 (9.8%) underwent mechanical ventilation. The total fatality rate among patients with COVID-19 was 14.9%. The median time of hospital stay was 9 days, yet it varied between groups of patients with certain outcomes (Table 1).

### 3.3. Liver Biochemistry Abnormalities

Liver test abnormalities were defined as the deviation of the following liver enzymes in serum: AST > 3 1 U/L, ALT > 35 U/L, GGT > 38 U/L, ALP > 150 U/L, ALP < 40 U/L, TBIL > 1.2 mg/dL, albumin < 3.5 g/L and albumin > 5.2 g/L. As COVID-19 is a new, emerging infectious disease, guidance or consensus on liver injury classifications is lacking.

Median values and percentages of abnormal laboratory tests at the time of hospital admission are presented in Table 2.

### 3.4. Association of Liver Biochemistry Abnormalities at Admission with COVID-19 Severity

In this section, we described the results obtained from testing models adjusted for comorbidities and potential risk factors (hypertension, diabetes, asthma, COPD, dementia, stroke/TIA in patient history, chronic kidney disease, myocardial infarction in patient history, heart failure, leukemia, lymphoma and solid malignant disease, smoking history and length of hospital stay), including single liver-related parameters and their double combinations. The sample in that part consisted of patients who had complete results of all laboratory tests considered in our research (*n* = 401).

Elevation of AST at admission was observed in 38.9% (*n* = 2109, Table 2). The AST model showed statistical significance for all endpoints at the significance level *p* < 0.001, but its role within the model was determined as significant only for COVID-19-related death (Table 3). Similar observations were made for AST models in combination with GGT, ALT, ALP, TBIL and albumin (Appendix A).

ICU: intensive care unit; AST: aspartate aminotransferase; ALT: alanine aminotransferase; GGT: gamma-glutamyltransferase; ALP: alkaline phosphatase; TBIL: total bilirubin; NLR: neutrophil-to-lymphocyte ratioAST combined with ALT revealed better predictive value for death than NLR, with *p* < 0.001 for the model and *p* = 0.002, OR 1.009, *p* = 0.03, OR 0.992 for the following parameters (Appendix A). 

Deviation in ALT, GGT and ALP levels were observed in 29.3%, 33.2% and 6.2% respectively (Table 2). Models that contain these parameters achieved a statistically significant predictive value *p* < 0.001 (Appendix A). However, in none of them, ALT, GGT or ALP were significant within the model, both alone (Table 3) or together with other characteristics (Appendix A).

TBIL elevation at admission was observed in a relatively small proportion of patients (*n* = 2109, Table 2), but it was associated significantly with higher mortality. The model for fatal outcome including TBIL reached a *p*-value at the level of<0.001, and TBIL significance was described as *p* = 0.004, OR 1.598. Interestingly, the model including TBIL and AST turned out to have a greater predictive value for death than the model containing NLR (Table 4), with *p*-value = 0.023, OR 1.471 for TBIL, and *p* = 0.038, OR 1.003 for AST (Appendix A).

Deviation in albumin levels on admission was presented by 21.1% of patients (*n* = 2109, Table 2). Hypoalbuminemia as a single test, as the only one of all investigated parameters, was statistically significantly associated with all three endpoints (Table 3). Remarkably, each of these models showed by far the highest predictive value for all the variables included (Appendix A). 

Albumin combined with each of the examined characteristics (AST, ALT, GGT, ALP and TBIL) produced a stronger predictive model for all endpoint tests than NLR, according to the AIC (Table 4). Albumin was the most important component examined in each of these tests, with a statistically significant effect on the final test result at the *p* < 0.0001 level (Appendix A).

ALP together with albumin performed better than albumin alone for ICU admission (Table 4), but a careful examination reveals that ALP was not statistically significant (*p* = 0.199, Appendix A). 

In the overview of models predicting the necessity of mechanical ventilation, we observed a similar distribution as in the case of ICU admission; however, for this particular outcome, the model using albumin without additional parameters works as well as the model combining albumin with ALP (Table 4).

The albumin-TBIL model (*p* < 0.001, Appendix A) and albumin together with AST (*p* < 0.001, Appendix A) were found to be the greatest predictors of fatal outcome (Table 4).

TBIL and albumin were the most important covariates in the model (*p* = 0.003, OR 1.705, *p* = 0.001, OR 0.228, respectively), outweighing the potential of other comorbidities (Appendix A). In the model combining albumin and AST, the predictive value of both factors within the model was statistically significant (*p* < 0.001, OR 0.233, *p* = 0.009, OR 1.003, respectively, Appendix A). 

Models examining the combination of albumin with other parameters (ALT, GGT, ALP) for fatal outcomes also performed better than the model with NLR (Table 4), while the inclusion of GGT and ALP seems to lower the value of the model compared to the one considering albumin alone (Table 4, Appendix A). ALT in combination with albumin fares better than albumin; however, ALT does not reach statistical significance within the model (*p* = 0.09, OR 1.002, Appendix A).

The comorbidities and risk factors with a statistically significant (*p* < 0.05) impact on ICU admission and mechanical ventilation were chronic kidney disease (OR < 1) and length of hospital stay (OR > 1) (Appendix A).

Within models testing the influence of included factors on the risk of death due to COVID-19, the most important (*p* < 0.05) were hypertension (OR > 1), dementia (OR > 1), myocardial infarction in the past (OR > 1) and leukemia (OR > 1). The laboratory parameters that always achieved greater statistical significance within the models for this endpoint were NLR and albumin (Appendix A).

### 3.5. Best Predictive Models

In this section, we present models with the highest predictive value for severe COVID-19, selected on the basis of *n* = 401 sample tests and examined on the entire population of patients included in the study (*n* = 2109), while excluding those with missing desired laboratory data. This means that samples tested in each model consisted of slightly different groups of patients, which is the reason for variation in statistical significance analyzed parameters, comorbidities and risk factors (hypertension, diabetes, asthma, COPD, dementia, stroke/TIA in patient history, chronic kidney disease, myocardial infarction in patient history, heart failure, leukemia, lymphoma and solid malignant disease, smoking history and length of hospital stay, Table 5, Table 6 and Table 7).

Within the models predicting the risk of ICU admission, which revealed the greatest impact on final outcome, the most important was hypoalbuminemia (OR < 0.3, *p* < 0.0001 for both models, Table 5). This time, despite the larger group of patients analyzed in the study, the model including ALP achieved better results than the model containing albumin as the only laboratory parameter, according to AIC. Moreover, the lack of statistical significance of ALP was confirmed. The better accuracy of the model containing ALP may result from the consideration of an additional parameter and a certain cooperation of ALP and albumin. Their reduced values may indicate cachexia and malnutrition.

Regarding mechanical ventilation as an endpoint, the reduced albumin level was assessed as the most powerful predictor in the models (Table 6).

The best model to predict COVID-19-related death became the one including albumin and TBIL, with albumin significance at the level of OR 0.229, *p* < 0.0001, and TBIL OR 1.657, *p* = 0.001 (Table 7). The model analyzing albumin and AST accompanied by other potential risk factors presented slightly worse according to AIC, but again both laboratory tests had influence (albumin OR 0.213, *p* < 0.0001, AST OR 1.003, *p* = 0.008, Table 7). Figure 1 shows the relationships between the best models for all endpoints.

## 4. Discussion

COVID-19, since it arose in Wuhan, China, has become a major public health concern all over the world. Early recognition of the severe course is an exclusive problem that is of vital importance for the management and use of medical facilities. In this situation, the search for a reliable combination of laboratory parameters that can be prognostic factors is definitely justified.

Lately, it has become evident that liver abnormalities are present in a relevant proportion of COVID-19 patients and correlate with worse outcomes [27]. The pattern of liver injury is predominantly hepatocellular or mixed-type and presents as aminotransferases and GGT elevation [21,22,23,24]. Nevertheless, we still do not have a sufficient explanation for this phenomenon. Direct viral effects on the liver have been discussed [12,13,14,17,28,29,30] since ACE2 is expressed in a subset of cholangiocytes, while expression in hepatocytes is low [12,13,14]. However, findings regarding the increased expression of ACE2 in liver cells due to hypoxia and inflammatory conditions should be mentioned [31,32]. Further potential mechanisms include liver damage through ACE2 receptors expressed in endothelial cells [33], but it has been confirmed that ACE2 is absent in sinusoidal endothelial cells [17]. The contribution of the CD47 receptor and the interactions between the L-SIGN receptor and ACE2 are other theories that should be mentioned [15,16]. Considering aminotranses, we need to note that the source of the elevated AST may be not the liver but the muscle or cardiac injury, which was suggested previously [34]. A study performed by Bloom et al. showed that there is moderate correlation between muscle injury markers such as creatine kinase, lactate dehydrogenase and AST elevation [35]. On the other hand, drug-induced liver injury is a possible contributing factor, based on recent studies. In the present research, we used only baseline parameters for prognostication, which allowed us to exclude the hepatotoxic potential of a wide array of drugs (e.g., redemsivir) used to treat COVID-19, mostly in off-label fashion. However, since many patients have chronic diseases, it is likely that chronic medications are taken. Therefore, in our multivariate models, we considered comorbidities, assessed their impact on the final value of the model and compared them with the influence of considered laboratory tests.

At the time of hospital admission, patients presented with liver biochemistry abnormalities 38.9% of AST, 29.3% of ALT, 33.2% of GGT, 21.1% of albumin, 6.7% of TBIL and 6.2% of ALP (Table 2).

The increased values of AST and ALT have already been widely commented on in the literature.

Lei et al. [9] reported a connection between liver injury based on markers of hepatic injury and inpatient mortality, specifically an association between AST abnormality and risk of death during hospitalization. Ramachandran et al. [36] claimed that elevated AST or ALT levels among hospitalized COVID-19 patients were associated with higher rates of mechanical ventilation but were not significant independent predictors of more severe disease. Pazgan-Simon et al. presented a rife elevation of ALT and AST at baseline, with no correlation with higher mortality [22]. In our cohort, AST was the most frequently elevated liver-related parameter, consistent with previously reported results. It has a significant predictive value for death in the course of COVID-19, but in the studied model, the presence of myocardial infarction in the patient’s history was more statistically significant (*p* = 0.008, OR 1.006 vs. *p* = 0.003, OR 2.805, respectively). That could bring us to the conclusion that AST may be elevated due to extrahepatic causes, such as exacerbation of previously existing coronary artery disease (CAD) already described in COVID-19 patients [37] or be the result of liver damage caused by chronic statin use due to CAD. AST in combination with TBIL outperformed the model with NLR, according to the AIC, but in this case, again, the most important factor in the model was myocardial infarction in the past (*p* = 0.003, OR 1.446 vs. *p* = 0.038, OR 1.0004 for AST, *p* = 0.023, OR 1.068 for TBIL). Nevertheless, it should be noted that elevated TBIL and AST levels on admission are independently associated with in-hospital death in patients with COVID-19.

ALT elevation is predominantly observed in hepatocellular injuries. Viral hepatitis classically leads to an ALT variation, which has also been reported in many studies focused on COVID-19 as more common than ALP or TBIL elevation but less common than AST, which does not match the viral pattern [21,24,35,38,39,40]. Bloom et al. [35] proved a strong correlation between AST and ALT, which may suggest that real hepatocellular injury is a predominant source of aminotransferase elevation. That conclusion is in line with the theory of cytokine-mediated injury or hypoxia proposed not only for COVID-19 [17,35,41] but also for influenza [42]. In our study, elevated ALT was observed in 29.3% of patients at baseline, which is comparable to the previously mentioned papers. In each model, ALT was not independently associated with the final outcome, besides combination with AST predicting death, in which it achieved significance at the level of *p* = 0.026, OR 0.986. This finding may support the theory that AST has a source other than the liver and that deviations in AST and ALT in COVID-19 patients are unrelated.

Hypoalbuminemia is a typical trait of all severely ill patients and is often related to inflammatory diseases, mainly due to increased vascular permeability (with increased albumin distribution volume) and a shortened albumin half-life. A low albumin level is a negative prognostic indication in hyperinflammatory conditions such as trauma, shock or infection and has been linked to poor outcomes and a shorter life expectancy [43]. The processes causing hypoalbuminemia in COVID-19 have not yet been fully investigated and described. Albumin is produced in the liver, but it has been subject to dispute as to whether its insufficiency in COVID-19 could be caused by liver dysfunction. According to the study performed by Huang et al. [44], this occurrence cannot be explained solely by liver damage as a result of hepatocellular dysfunction; nonetheless, they did not find any correlation between AST and ALT elevation and worse outcomes, which might be explained by population differences.

The literature often addresses hypoalbuminemia in severe COVID-19 [24,45,46], but the prognostic value of albumin is still underestimated. According to our research, lower albumin levels on admission can predict COVID-19 outcomes irrespective of most comorbidities and better than NLR. Our results emphasize the remarkable effectiveness of albumin as a predictor of intubation necessity. Models tested on a smaller group of patients *n* = 401, which included albumin, turned out to be a better indicator of COVID-19 severity than the NLR for all endpoints and, along with TBIL, the best model to predict the risk of death, which aligns with Weber and colleagues’ findings [24]. Similarly, hypoalbuminemia, in combination with variation in ALP, became the best combination to assess the risk of a patient’s admission to the ICU.

The albumin level confirmed its prognostic efficiency during testing the models on larger groups of patients (Table 5, Table 6 and Table 7).

Moreover, according to the results of our study, attention should be paid to the assessment of TBIL and AST levels, whose significance was confirmed during testing on larger samples (Table 7). Although some of the variables considered exceeded their importance, each of them turned out to be significant, so they can be independent predictors of death in the course of COVID-19, working better when assessed in combination with albumin.

The majority of the available data show that the cholestatic pattern of injury is less common in COVID-19 patients than AST and ALT, but mildly raised GGT levels can be seen in up to 50% of patients [7,21,22,23,24,35,38,39,40]. In our study, GGT, ALP and TBIL were varied in 33.2%, 6.2% and 6.7% of patients, respectively.

GGT elevation and its predictive value in terms of COVID-19 severity have been widely commented upon recently. Shao et al. reported that elevated GGT and CRP levels were associated with a longer hospital stay [47]. Kasapoglu and colleagues determined that elevated serum GGT levels, but not aminotransferases, at admission were associated with the increased risk of ICU admission and mortality [48], while Weber et al. claimed that AST, ALT, GGT and albumin correlated strongly with COVID-19-related death [24]. Our results show that, despite the fact that elevated GGT is frequently observed at baseline, it is not the best predictor for COVID-19 severity. Elevated ALP levels are rather rare among patients admitted to hospital for COVID-19, and worldwide prevalence reaches 4% [21,24,35]. In our study, abnormal ALP was observed in a similar percentage. Although uncommon, it should not be neglected. Da et al., in their study, assessed the effect of cholestasis, defined as the serum ALP level > 3x upper normal limits (UNL), on the mortality of COVID-19 patients [49]. Our research did not reveal any statistically significant effect of ALP on COVID-19 severity, neither in models with ALP alone nor in different combinations, tested both, on sample *n* = 401 or *n* = 459 (Table 3, Table 5 and Table 6). Consideration of an extra parameter, as well as a certain corporation of ALP and albumin, whose lower levels may suggest cachexia and malnutrition, could result in improved accuracy of the model comprising ALP (Table 4, Table 5 and Table 6).

Bilirubin, a natural end product of haeme catabolism, has long been recognized as a protective molecule with powerful antioxidant, anti-inflammatory, and other bioactivities [50]. Liu et al. [51] observed that a low percentage of COVID-19 patients had elevated bilirubin levels, and this group tended to have a worse outcome and a more severe illness. Furthermore, those with greater TBIL levels at admission had a higher mortality risk. Ding et al. [52] found that abnormal AST and direct bilirubin baseline and peak are associated with in-hospital fatal outcomes. Weber et al. [24] claimed that TBIL elevation was the most predictive single factor of COVID-19-related death. In our study, the most influential singular factor remained hypoalbuminemia, but hypoalbuminemia and TBIL elevation on admission became the best predictive model for in-hospital mortality due to COVID-19. Moreover, it is necessary to consider the predictive ability of the TBIL and AST combination, which value exceeded the model for NLR and the fact that the role of each parameter in the model also proves to have a significant impact on final fatal outcome.

Referring to the observed impact of comorbidities and potential risk factors, the opposite effect of chronic kidney disease on the risk of admission to the ICU and intubation was noted. In the analysis with the endpoint of death due to COVID-19, chronic kidney disease did not show statistical significance; however, the OR also took a value < 1 (Table 7, Appendix A). This observation is in contrast to the currently available data [53,54]. The reason for this unexpected remark may be the sample size and the precise group of patients who had performed admission tests. Similar observations were made for smoking history (Table 5 and Table 6), contrary to popular claims [37,55]. Dementia was shown to be adversely linked with the risk of ICU admission and invasive respiratory support in our cohort, but not with the risk of mortality due to SARS-CoV-2 infection (OR > 1, Table 7). Although the lack of significance may be due to the small size of the research group, the tendency of dementia to indicate COVID-19-related death is visible and consistent with previous observations [56]. CVD and type 2 diabetes mellitus are well-known risk factors for the severe course of COVID-19, and our survey for these comorbidities is compatible [57,58]. Leukemia was also significantly associated with mortality, in line with earlier considerations [59], and our group had a strong influence on the risk of admission to the ICU.

In this research, patients with chronic liver diseases known as chronic hepatitis, cirrhosis and fatty liver were excluded from the analysis. This allowed us for an objective assessment of the liver function of COVID-19 patients on admission, but at the same time made it impossible to investigate in depth the impact of these comorbidities on the severe course of the disease. However, this influence was reported previously by Galiero et al. in a large multicenter study [60].

Our study had several limitations. First, due to the retrospective nature of the study, the results of the laboratory tests on admission were incomplete. This caused a constraint of the sample to *n* = 401 during the creation of adequate and reliable models that were later implemented on the entire group of patients included in the study (*n* = 2109). Second, we investigated test results obtained at admission to the hospital, which caused an inability to analyze further fluctuations in the parameters in the course of the disease. Moreover, in our study, we did not include the international normalized ratio (INR), which could be a useful tool to assess liver function and to complete our conclusions. In our study, we excluded patients with chronic liver diseases, which did not allow us to evaluate how SARS-CoV-2 infection affects disease progression or how the deterioration of liver functioning in these cases might affect the course of COVID-19. Finally, because the study was single-center, a certain selectivity of the analyzed group may have resulted in bias. Therefore, prospective, long-term studies are necessary to verify the validity of our observations and conclusions.

## 5. Conclusions

In our study, we attempted to find predictive factors for the severe course of SARS-CoV-2 infection, which is important for risk classification, optimizing hospital resource redistribution, and guiding healthcare strategies.

Our observations emphasize that hypoalbuminemia is a strong predictor of severe COVID-19 course and, in combination with AST or TBIL, has a remarkable association with mortality. This is the most important finding for clinicians because the assessment of the albumin, AST and TBIL levels at admission is an inexpensive, quick test that can be performed in most patients and allows for a more accurate prognosis in the course of SARS-CoV-2 infection.

Moreover, based on our results, we suspect that the liver may not be the source of the elevated AST level, which is often observed in COVID-19 patients. Deviated AST on admission should draw our attention to the feasible risk of exacerbation of CVD and cannot be underestimated; however, further research is required to fully explore this assumption.

Despite the presence of abundant ACE-2 receptors in cholangiocytes, the parameters associated with cholestasis did not appear to be useful in the early risk assessment of severe COVID-19. GGT and ALP were not found to be predictors of worse outcomes; nonetheless, elevated TBIL levels, although rare, are a significant predictor of death in the course of SARS-CoV-2 infection.

Overall, the early determination of biochemical parameters in COVID-19 patients can provide important prognostic information. Decreased albumin, high AST and TBIL levels should be alarming as potentially associated with a severe course of the disease.

## Figures and Tables

**Figure 1 jcm-11-04490-f001:**
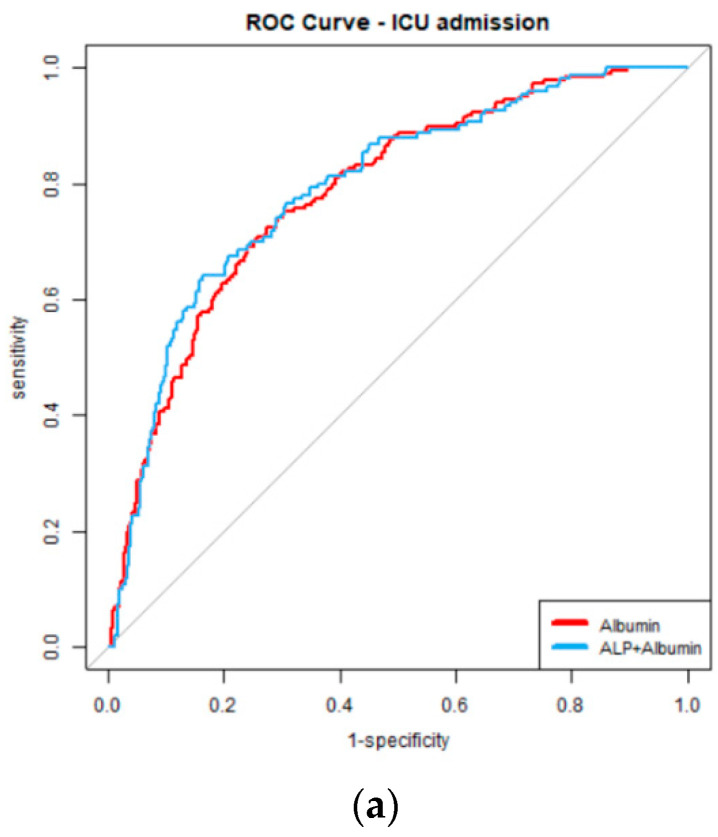
ROC curves presenting relationships between best models for endpoints (**a**) ICU admission, (**b**) mechanical ventilation, and (**c**) fatal outcome. The models, apart from the laboratory parameters, were adjusted for hypertension, diabetes, asthma, COPD, dementia, stroke/TIA in patient history, chronic kidney disease, myocardial infarction in patient history, heart failure, leukemia, lymphoma and solid malignant disease, smoking history and length of hospital stay.

**Table 1 jcm-11-04490-t001:** Baseline demographic data and comorbidities of 2184 patients hospitalized due to COVID-19.

	All Patients	ICU Admission	*p*-Value	Mechanical Ventilation	*p*-Value	Fatal Outcome	*p*-Value
	*n* = 2184	*n* = 214		*n* = 215		*n* = 326	
		% 9.8		% 9.8		% 14.9	
Age, median	64 (46–73)	64 (52–70)		65 (54–71)		72.5 (65–84)	
Gender, n (%)			<0.001		<0.001		<0.001
Female	1102 (50.4%)	71 (6.4%)		70 (6.4%)		129 (11.7%)	
Male	1082 (49.5%)	143 (13.2%)		145 (13.4%)		197 (18.2%)	
Underlying comorbidities, n (%)							
Hypertension	1022 (46.8%)	125 (5.7%)	<0.001	130 (5.95%)	<0.001	219 (10%)	<0.001
Diabetes	516 (23.6%)	75 (3.4%)	<0.001	79 (3.6%)	<0.001	122 (5.6%)	<0.001
Asthma	85 (3.9%)	12 (0.6%)	0.172	11 (0.5%)	0.328	14 (0.6%)	0.684
COPD	75 (3.4%)	5 (0.2%)	0.353	6 (0.3%)	0.585	18 (0.8%)	0.025
Dementia	132 (6%)	3 (0.1%)	0.003	6 (0.3%)	0.035	49 (2.2%)	<0.001
Stroke/TIA in patient history	164 (7.5)	13 (0.6%)	0.402	16 (0.7%)	0.969	35 (1.6%)	0.017
Chronic kidney disease	231 (10.6)	21 (1%)	0.702	22 (1%)	0.863	72 (3.3%)	<0.001
Myocardial infraction in patient history	191 (8.7%)	28 (1.3%)	0.018	32 (1.5%)	<0.001	74 (3.4%)	<0.001
Heart failure	255 (11.7%)	33 (1.5%)	0.072	33 (1.5%)	0.077	89 (4.1%)	<0.001
Chronic liver disease	74 (3.4%)	7 (0.3%)	0.802	5 (0.2%)	0.83	12 (0.6%)	0.508
Solid malignant disease	151 (6.9%)	10 (0.5%)	0.098	10 (0.5%)	0.096	42 (1.9%)	<0.001
Leukemia	19 (0.9%)	6 (0.3%)	0.001	4 (0.2%)	0.1	8 (0.4%)	0.001
Lymphoma	18 (0.8%)	0 (0%)	0.200	0 (0%)	0.2	6 (0.3%)	0.03
AIDS	2 (0.1%)	0 (0%)	0.641	0 (0%)	0.640	0 (0%)	0.553
Smoking status, n (%)	193 (8.8%)	15 (0.7%)	0.606	15 (0.7%)	0.606	41 (1.9%)	0.004
Active smoker	117 (5.4%)	6 (0.3%)		6 (0.3%)		11 (0.5%)	
Former smoker	76 (3.5%)	9 (0.4%)		9 (0.4%)		30 (1.4%)	
Length of hospital stay days, median	9 (2–16)	18 (9–27)		16 (6–26)		13 (5–21)	

Values for continuous variables were showed as or median (Q25–Q75). ICU: intensive care unit; COPD: chronic obstructive pulmonary disease; TIA: transient ischemic attack; AIDS: active acquired immunodeficiency syndrome.

**Table 2 jcm-11-04490-t002:** Labolatory parameters of patients on admission *n* = 2109.

	Median (IQR)	Abnormal (%)
AST (0–31 U/L)	37 (24–62)	38.9
ALT (0–35 U/L)	29 (18–50)	29.3
GGT (0–38 U/L)	42 (24–83)	33.2
ALP (40–150 U/L)	65 (51–95)	6.2
TBIL (0.2–1.2 mg/dL)	0.6 (0.5–0.8)	6.7
Albumin (3.5–5.2 g/L)	3.1 (2.7–3.5)	21.1
Neu (2.5 × 10^3^–6 × 10^3^/mm^3^) (×10^3^)	5.5 (3.5–8.4)	34.0
Lym (1.5 × 10^3^–3.5 × 10^3^/mm^3^) (×10^3^)	1.0 (0.7–1.4)	46.7
NLR	5.9 (3.1–10.8)	

IQR: Interquartial Range; AST: aspartate aminotransferase; ALT: alanine aminotransferase; GGT: gamma-glutamyltransferase; ALP: alkaline phosphatase; TBIL: total bilirubin; Neu: neutrophil; Lym: lymphocyte; NLR: neutrophil-to-lymphocyte ratio.

**Table 3 jcm-11-04490-t003:** The role of liver-related laboratory parameters in models containing potential risk factors and their impact on outcome.

	ICU Admission			Mechanical Ventilation			Fatal Outcome		
	OR	95% CI	*p*-Value	OR	95% CI	*p*-Value	OR	95% CI	*p*-Value
Liver biochemistry abnormality									
AST	1.000	0.998–1.000	0.6	1.000	0.999–1.000	0.32	1.004	1.001–1.010	0.008
ALT	1.000	0.998–1.000	0.55	1.000	0.999–1.000	0.46	1.002	0.999–1.00	0.133
GGT	1.000	0.999–1.000	0.74	1.000	0.999–1.000	0.60	1.001	0.999–1.000	0.220
ALP	0.999	0.996–1.000	0.41	0.999	0.996–1.000	0.29	1.002	0.999–1.00	0.087
TBIL	0.965	0.698–1.270	0.81	0.985	0.717–1.300	0.92	1.598	1.188–2.250	0.004
albumin	0.281	0.172–0.450	<0.001	0.247	0.148–0.400	<0.001	0.233	0.143–0.370	<0.001
NLR	1.033	1.015–1.060	0.001	1.032	1.014–1.050	0.001	1.031	1.011–1.060	0.006

Stated are the OR (Odds Ratio), 95% CI (Confidence Interval) and *p*-values for laboratory parameters, each in single model adjusted for comorbidities. A full model description is available in Appendix A.

**Table 4 jcm-11-04490-t004:** Comparison of the models with single parameters and their double combinations adjusted for comorbidities and potential risk factors according to AIC for individual endpoints.

	ICU-Admission			Mechanical Ventilation			Fatal Outcome	
	Model	AIC		Model	AIC		Model	AIC
1	albumin+ALP	439.0	1	albumin	426.2	1	albumin+TBIL	455.1
2	albumin	439.2	2	albumin+ALP	426.2	2	albumin+AST	456.8
3	albumin+ALT	440.7	3	albumin+ALT	427.3	3	albumin+ALT	462.1
4	albumin+TBIL	441.0	4	albumin+AST	427.5	4	albumin	464.1
5	albumin+AST	441.2	5	albumin+TBIL	428.0	5	albumin+GGT	464.7
6	albumin+GGT	441.2	6	albumin+GGT	428.1	6	albumin+ALP	464.7
7	NLR	456.3	7	NLR	448.4	7	AST+ALT	493.2
8	ALP	469.8	8	AST	461.1	8	AST+TBIL	493.5
9	ALT	470.4	9	AST+ALP	461.5	9	NLR	497.0
10	AST	470.4	10	ALP	461.6	10	AST	497.2
11	GGT	470.5	11	ALT	461.6	11	TBIL+GGT	497.2
12	TBIL	470.6	12	GGT	461.9	12	TBIL	497.5
13	AST+ALP	470.9	13	TBIL	462.1	13	TBIL+ALT	497.7
14	ALP+GGT	470.9	14	ALP+GGT	462.4	14	AST+ALP	498.7
15	ALP+ALT	471.0	15	ALP+ALT	462.6	15	AST+GGT	499.1
16	ALP+TBIL	471.6	16	AST+TBIL	462.8	16	ALP+TBIL	499.5
17	TBIL+GGT	472.1	17	AST+GGT	463.1	17	ALP+ALT	504.6
18	AST+TBIL	472.1	18	AST+ALT	463.1	18	ALP	504.6
19	TBIL+ALT	472.2	19	ALP+TBIL	463.3	19	ALT	504.7
20	ALT+GGT	472.2	20	TBIL+GGT	463.4	20	ALT+GGT	505.8
21	ALT+AST	472.3	21	ALT+GGT	463.4	21	GGT	506.4
22	AST+GGT	472.3	22	TBIL+ALT	463.5	22	TBIL+GGT	506.5

AIC: Akaike Information Criterion; ICU: intensive care unit; AST: aspartate aminotransferase; ALT: alanine aminotransferase; GGT: gamma-glutamyltransferase; ALP: alkaline phosphatase; TBIL: total bilirubin; NLR: neutrophil-to-lymphocyte ratio.

**Table 5 jcm-11-04490-t005:** The best models to predict ICU admission in COVID-19 tested on larger groups of patients.

ICU Admission								
	**OR**	**95% CI**		***p*-Value**	**Sample Size n**	**NG**	**LR** ***p*-Value**	**AIC**
**Variables**		**25%**	**75%**					
**Model: albumin+ALP**	36.810	8.245	177.080	<0.0001	459	0.292	<0.0001	518.2
ALP	0.997	0.994	1.000	0.101				
albumin	0.277	0.176	0.420	<0.0001				
Hypertension	1.103	0.675	1.810	0.694				
Diabetes-1	0.243	0.012	1.570	0.207				
Diabetes-2	1.504	0.824	2.740	0.182				
Diabetes-3	1.881	0.754	4.650	0.170				
Diabetes-4	2.234	0.588	8.410	0.228				
Diabetes-5	<0.001	<0.001	<0.001	0.997				
Asthma	1.837	0.579	5.700	0.290				
COPD	0.236	0.033	1.050	0.088				
Dementia	0.061	0.009	0.230	<0.001				
Stroke/TIA in patient history	0.650	0.257	1.540	0.341				
Chronic kiedney disease	0.279	0.138	0.540	<0.001				
Active smoker	0.706	0.192	2.300	0.576				
Former smoker	0.279	0.069	0.910	0.047				
Myocardial infraction in patient history	1.704	0.802	3.630	0.164				
Heart failure	0.959	0.473	1.910	0.905				
Leukemia	4.327	0.946	21.070	0.058				
Solid malignant disease without metastases	0.863	0.317	2.260	0.767				
Solid malignant disease with metastases	<0.001	<0.001	>1000	0.988				
Lymphoma	<0.001	<0.001	>1000	0.989				
Lenght of hospital stay	1.007	0.993	1.020	0.341				
	**OR**	**95% CI**		***p*-Value**	**Sample Size n**	**NG**	**LR** ***p*- Value**	**AIC**
**Variables**		**25%**	**75%**					
**Model: albumin**	21.583	6.137	79.200	<0.0001	621	0.276	<0.0001	666
albumin	0.295	0.200	0.430	<0.0001				
Hypertension	1.055	0.685	1.630	0.808				
Diabetes-1	0.224	0.012	1.300	0.168				
Diabetes-2	1.751	1.038	2.950	0.035				
Diabetes-3	1.782	0.809	3.870	0.146				
Diabetes-4	2.423	0.730	8.030	0.142				
Diabetes-5	<0.001	<0.001	>1000	0.994				
Asthma	1.749	0.708	4.260	0.218				
COPD	0.514	0.142	1.610	0.274				
Dementia	0.068	0.015	0.210	<0.001				
Stroke/TIA in patient history	0.748	0.329	1.600	0.468				
Chronic kiedney disease	0.263	0.139	0.470	<0.001				
Active smoker	0.773	0.268	2.070	0.618				
Former smoker	0.352	0.105	1.000	0.065				
Myocardial infraction in patient history	1.495	0.780	2.840	0.221				
Heart failure	0.861	0.467	1.560	0.625				
Leukemia	4.333	1.092	17.900	0.036				
Solid malignant disease without metastases	0.532	0.217	1.200	0.144				
Solid malignant disease with metastases	<0.001	<0.001	>1000	0.984				
Lymphoma	<0.001	<0.001	>1000	0.987				
Lenght of hospital stay	1.006	0.994	1.020	0.333				

OR: Odds Ratio; 95% CI: Confidence Interval; NG: Nagelkerke pseudo R2; LR: Likelihood Ratio; AIC: Akaike Information Criterion; ICU: intensive care unit; ALP: alkaline phosphatase; Diabetes-1:diabetes mellitus type 1, including LADA; Diabates-2: diabetes mellitus type 2 treated with oral medications; Diabetes-3: diabetes mellitus type 2 treated with insulin; Diabtes-4: prediabetes; Diabtes-5: gestational diabetes; COPD: chronic obstructive pulmonary disease; TIA: transient ischemic attack.

**Table 6 jcm-11-04490-t006:** The best models to predict nesscesity of mechanical ventilation in COVID-19 tested on larger groups of patients.

Mechanical Ventilation								
	**OR**	**95% CI**		***p*-Value**	**Sample Size n**	**NG**	**LR** ***p*-Value**	**AIC**
**Variables**		**25%**	**75%**					
**Model: albumin+ALP**	50.933	11.113	253.280	<0.0001	459	0.298	<0.0001	507.8
ALP	0.998	0.994	1.000	0.121				
albumin	0.244	0.154	0.380	<0.0001				
Hypertension	1.334	0.810	2.210	0.259				
Diabetes-1	0.244	0.012	1.590	0.209				
Diabetes-2	1.331	0.723	2.440	0.356				
Diabetes-3	2.102	0.828	5.330	0.115				
Diabetes-4	2.232	0.582	8.530	0.233				
Diabetes-5	<0.001	<0.001	<0.001	0.997				
Asthma	1.342	0.404	4.180	0.616				
COPD	0.410	0.080	1.610	0.230				
Dementia	0.057	0.008	0.220	<0.001				
Stroke/TIA in patient history	0.735	0.287	1.760	0.502				
Chronic kiedney disease	0.270	0.131	0.530	<0.001				
Active smoker	0.765	0.208	2.500	0.668				
Former smoker	0.211	0.043	0.760	0.030				
Myocardial infraction in patient history	1.622	0.751	3.500	0.216				
Heart failure	0.649	0.309	1.320	0.241				
Leukemia	1.515	0.266	7.180	0.611				
Solid malignant disease without metastases	0.739	0.260	1.970	0.555				
Solid malignant disease with metastases	<0.001	<0.001	>1000	0.987				
Lymphoma	<0.001	<0.001	>1000	0.988				
Lenght of hospital stay	1.005	0.991	1.020	0.497				
	**OR**	**95% CI**		***p*-Value**	**Sample Size n**	**NG**	**LR** ***p*-Value**	**AIC**
**Variables**		**25%**	**75%**					
**Model: albumin**	23.245	6.524	86.500	<0.0001	621	0.266	<0.0001	657.3
albumin	0.284	0.192	0.410	<0.0001				
Hypertension	1.228	0.794	1.900	0.356				
Diabetes-1	0.233	0.012	1.360	0.180				
Diabetes-2	1.677	0.990	2.830	0.053				
Diabetes-3	2.286	1.043	4.980	0.037				
Diabetes-4	2.721	0.826	8.980	0.095				
Diabetes-5	<0.001	<0.001	>1000	0.994				
Asthma	1.572	0.627	3.820	0.322				
COPD	0.512	0.142	1.610	0.273				
Dementia	0.072	0.016	0.220	<0.001				
Stroke/TIA in patient history	0.800	0.351	1.720	0.577				
Chronic kiedney disease	0.294	0.157	0.530	<0.001				
Active smoker	0.731	0.243	2.000	0.556				
Former smoker	0.356	0.104	1.030	0.073				
Myocardial infraction in patient history	1.686	0.880	3.210	0.112				
Heart failure	0.647	0.345	1.190	0.167				
Leukemia	1.770	0.399	7.030	0.426				
Solid malignant disease without metastases	0.488	0.191	1.120	0.108				
Solid malignant disease with metastases	<0.001	<0.001	>1000	0.984				
Lymphoma	<0.001	<0.001	>1000	0.987				
Lenght of hospital stay	1.000	0.989	1.010	0.953				

OR: Odds Ratio; 95% CI: Confidence Interval; NG: Nagelkerke pseudo R2; LR: Likelihood Ratio; AIC: Akaike Information Criterion; ALP: alkaline phosphatase; Diabetes-1:diabetes mellitus type 1, including LADA; Diabates-2: diabetes mellitus type 2 treated with oral medications; Diabetes-3: diabetes mellitus type 2 treated with insulin; Diabtes-4: prediabetes; Diabtes-5: gestational diabetes; COPD: chronic obstructive pulmonary disease; TIA: transient ischemic attack.

**Table 7 jcm-11-04490-t007:** The best models to predict fatal outcomes in COVID-19 tested on larger groups of patients.

Fatal Outcome								
	**OR**	**95% CI**		***p*-Value**	**Sample Size n**	**NG**	**LR** ***p*-Value**	**AIC**
**Variables**		**25%**	**75%**					
Model: albumin+TBIL	26.886	7.227	105.600	<0.0001	582	0.291	<0.0001	645
TBIL	1.657	1.249	2.290	0.001				
albumin	0.229	0.151	0.340	<0.0001				
Hypertension	1.538	0.982	2.420	0.061				
Diabetes-1	0.800	0.159	3.190	0.764				
Diabetes-2	1.814	1.071	3.070	0.026				
Diabetes-3	1.723	0.819	3.580	0.146				
Diabetes-4	3.292	0.895	11.440	0.063				
Diabetes-5	<0.001	<0.001	>1000	0.984				
Asthma	1.191	0.432	3.040	0.723				
COPD	0.806	0.259	2.360	0.700				
Dementia	1.473	0.709	3.070	0.298				
Stroke/TIA in patient history	0.916	0.449	1.820	0.804				
Chronic kiedney disease	0.614	0.354	1.050	0.077				
Active smoker	1.110	0.405	2.880	0.835				
Former smoker	0.818	0.283	2.200	0.699				
Myocardial infraction in patient history	2.904	1.609	5.310	<0.001				
Heart failure	1.313	0.750	2.290	0.337				
Leukemia	6.608	1.617	30.450	0.010				
Solid malignant disease without metastases	1.017	0.443	2.240	0.967				
Solid malignant disease with metastases	0.286	0.065	1.010	0.068				
Lymphoma	1.895	0.338	8.780	0.431				
Lenght of hospital stay	0.974	0.960	0.990	<0.001				
	**OR**	**CI 95%**		***p*-Value**	**Sample Size n**	**NG**	**LR** ***p*-Value**	**AIC**
**Variables**		**25%**	**75%**					
Model: albumin+AST	38.776	10.422	152.550	<0.0001	603	0.305	<0.0001	661.5
albumin	0.213	0.140	0.310	<0.0001				
AST	1.003	1.001	1.010	0.008				
Hypertension	1.573	1.009	2.460	0.046				
Diabetes-1	1.021	0.276	3.450	0.973				
Diabetes-2	1.685	1.007	2.810	0.046				
Diabetes-3	1.533	0.724	3.200	0.258				
Diabetes-4	3.133	0.838	11.120	0.079				
Diabetes-5	<0.001	<0.001	>1000	0.980				
Asthma	1.011	0.363	2.600	0.983				
COPD	0.823	0.267	2.390	0.725				
Dementia	1.550	0.749	3.220	0.237				
Stroke/TIA in patient history	0.925	0.454	1.840	0.827				
Chronic kiedney disease	0.717	0.414	1.220	0.228				
Active smoker	1.267	0.484	3.190	0.621				
Former smoker	0.693	0.250	1.780	0.461				
Myocardial infraction in patient history	2.751	1.542	4.970	0.001				
Heart failure	1.520	0.882	2.610	0.129				
Leukemia	6.975	1.720	32.450	0.008				
Solid malignant disease without metastases	1.250	0.566	2.690	0.572				
Solid malignant disease with metastases	0.537	0.159	1.640	0.289				
Lymphoma	1.829	0.331	8.470	0.454				
Lenght of hospital stay	0.975	0.962	0.990	< 0.001				

OR: Odds Ratio; 95% CI: Confidence Interval; NG: Nagelkerke pseudo R2; LR: Likelihood Ratio; AIC: Akaike Information Criterion; ICU: intensive care unit; TBIL: total bilirubin; AST: aspartate aminotransferase; Diabetes-1:diabetes mellitus type 1, including LADA; Diabates-2: diabetes mellitus type 2 treated with oral medications; Diabetes-3: diabetes mellitus type 2 treated with insulin; Diabtes-4: prediabetes; Diabtes-5: gestational diabetes; COPD: chronic obstructive pulmonary disease; TIA: transient ischemic attack.

## Data Availability

The datasets used and/or analyzed during the current study are available from the corresponding author upon reasonable request.

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
