# Peer review of "Liver Function Tests in COVID-19: Assessment of the Actual Prognostic Value"

_jcm, 2022, doi:10.3390/jcm11154490_

Round 1
Reviewer 1 Report
1. paper is too long needs to be shortened
2. the first paragraph of introduction is already obsolete, please delete and start describing gastrointestinal and hepatic manifestations of COVID instead
3. Introduction- line 81; Patients with NAFLD have particular susceptibility to COVID related liver abnormalities ( please see : https://www.ncbi.nlm.nih.gov/pmc/articles/PMC8087474/ and https://www.mdpi.com/1648-9144/57/10/1057)
4. What was the rationale not to include INR? It is a marker of liver synthetic function..
5. What is the rationale not to control for BMI in stat analysis? malnutrition with low albumin and BMI would be expected to be the risk factor
Author Response
We would like to thank the reviewer very much for the constructive criticism and the comments. Thank you very much. Regarding the concrete suggestions we answered them point-by-point as follows:
- “paper is too long needs to be shortened”
Dear Sir/Madam, thank You very much for Your suggestion which is in line with the opinion of the other reviewer. We managed to short our paper, especially the Results part ,which was overpacked with statistical data. We tried to do this without losing the important information. We hope now the article will be more suitable for the readers.
- “the first paragraph of introduction is already obsolete, please delete and start describing gastrointestinal and hepatic manifestations of COVID instead”
Dear Sir/Madam thank You very much for that opinion. We agree that there is no point in duplicating commonly known issues in such a long article. Therefore, we changed the first paragraph, leaving only one sentence of the introduction to the topic, and went straight to the description of possible manifestations of COVID-19, focusing on gastrointestinal manifestations.
- “Introduction- line 81; Patients with NAFLD have particular susceptibility to COVID related liver abnormalities (please see : https://www.ncbi.nlm.nih.gov/pmc/articles/PMC8087474/ and https://www.mdpi.com/1648-9144/57/10/1057)”
Dear Sir/Madam, we are grateful for sharing with us Your knowledge about links between COVID-19 and NAFLD. It helped us to improve part of introduction discussing the possible pathogenesis of changes in liver functioning during the SARS-CoV-2 infection. We also added proper references. Thank You again, we hope that we have met Your expectations.
- “What was the rationale not to include INR? It is a marker of liver synthetic function.”
Dear Sir/Madam, thank You very much for that suggestion. It is right that INR could be a great marker of liver function, especially analyzed together with albumin level. Conducting of the study assessing both of these parameters as markers of liver failure is definitely needed and important to consider in the future. However our analyses, contain a lot of variables already. So we decided that adding another variable will not increase the value of our research very much, but will complicate the calculations. In addition, the variability of INR is subject to many influences, as is the level of albumin, and determining the causes of these changes in the group of COVID-19 patients requires separate attention and careful analysis. Considering the above, we have hope that our failure to analyze INR will be understood and will not subtract the value of our work. Also we added separate information about not considering INR in our study limitations section.
- “What is the rationale not to control for BMI in stat analysis? malnutrition with low albumin and BMI would be expected to be the risk factor.”
Dear Sir / Madam, thank You very much for that comment. We agree that BMI could be an important supplement to our analysis, especially in the context of malnutrition. However, it should be noted that obese patients may also be malnourished, and the objectification of data in the case of sarcopenic obesity would require a validated technique of body composition measurement (e.g. electrical bioimpedance method). Therefore, we believe that in order to reliably assess malnutrition in COVID-19 patients, the Global Leadership Conversation Addressing Malnutrition 2018 criteria should be used in prospective study. It would be very interesting and could certainly be the subject of another article focused on malnutrition in COVID-19 patients.
Dear reviewer,
thank you very much for this detailed, accurate and extremely helpful review.
Thank you.
Reviewer 2 Report
This study, although retrospective and monocentric, is interesting and quite well written. This review raises only a few comments.
1- The abstract is missing the numerical data relating to the study. In particular, the number of patients recruited and the main results of the study are missing. Please add this information to the abstract.
2- The authors focused in this study on the effects on liver markers likely due to COVID. Therefore, patients with chronic liver diseases known as chronic hepatitis, cirrhosis and fatty liver were excluded from the analysis. This methodological choice did not allow to evaluate the impact of chronic liver disease on clinical outcomes in hospitalized patients for COVID, as has been documented in the other studies (PLoS One. 2020; 15(12): e0243700.doi: 10.1371/journal.pone.0243700). This issue and the previous reference should be added to the discussion.
3- The limitations section of the study should be broader.
Author Response
We would like to thank the reviewer very much for all suggestions and comments. They are incredible important for us. Regarding the concrete suggestions we answered them point-by-point as follows:
- “This study, although retrospective and monocentric, is interesting and quite well written. This review raises only a few comments.”
Dear Sir / Madam, we are very grateful for Your comment. We have tried to present interesting results, despite the limitations of our study. Our aim was to raise important issues that have clinical implications for management in COVID-19 patients.
- “1- The abstract is missing the numerical data relating to the study. In particular, the number of patients recruited and the main results of the study are missing. Please add this information to the abstract.”
Dear Sir/Madam, thank You for that suggestion. We added numerical data to the abstract: sample size and main results. We hope that now our abstract is complete.
- “2- The authors focused in this study on the effects on liver markers likely due to COVID. Therefore, patients with chronic liver diseases known as chronic hepatitis, cirrhosis and fatty liver were excluded from the analysis. This methodological choice did not allow to evaluate the impact of chronic liver disease on clinical outcomes in hospitalized patients for COVID, as has been documented in the other studies (PLoS One. 2020; 15(12): e0243700.doi: 10.1371/journal.pone.0243700). This issue and the previous reference should be added to the discussion.”
Dear Sir/Madam, as You mentioned our study was mainly focused on liver abnormalities associated with COVID-19 and that was the reason of excluding patients with preexisting chronic liver disease (CLD). Therefore, the influence of CLD on COVID-19 severity was emphasized in numerous studies and considering it is important to understand complexity of the problem of pathogenesis of SARS-CoV-2 virus influence on the liver. These are reasons why we supplemented the discussion with the desired information, and added the suggested article to the bibliography. Furthermore we emphasized that methodological choice as a limitation of our study. Hopefully now our analysis is much more comprehensive.
- “3- The limitations section of the study should be broader”
Dear Sir/Madame, we considered Your comment and add some limitations based on Yours suggestions and other reviewers. We claimed that exclusion of patients with chronic liver disease limited our ability to assess the influence of the SARS-CoV-2 infection on the liver function. We have also added information that considering INR as an additional parameter in our models could be valuable. We hope that broader limitations will inspire our readers to design study that will complement our understanding of liver damage in COVID-19.
Dear reviewer, thank you very much for this accurate and helpful review. Thank you.
Reviewer 3 Report
The article by Tokarczyk U, Kaliszewski K et al. aims to assess the prognostic value of liver function tests with COVID-19 severity and outcomes. A number of studies have already been published regarding this issue, so the study is not original in this matter. Also, the number of patients included is not very high. The manuscript is very difficult to read, as the emphasis of the paper has been put on statistical methods and analyses, and not on the clinical significance of the findings. The English language requires improvements, because in some parts of the manuscript the style impairs the quality of the manuscript, as sentences are difficult to understand.
The manuscript should be rewritten in the manner to be shorter, more informative for the clinicians and easier to understand and follow.
Author Response
Dear Reviewer, thank You very much for all Your advices and comments. Regarding the concrete suggestions we answered them point-by-point as follows:
- “The article by Tokarczyk U, Kaliszewski K et al. aims to assess the prognostic value of liver function tests with COVID-19 severity and outcomes. A number of studies have already been published regarding this issue, so the study is not original in this matter. “
Dear Sir / Madame, thank You very much for Your opinion. In our study we aimed to raise an important issue regarding management of COVID-19 patients. We are aware that many publications have already raise the topic of liver enzymes abnormalities in the course of SARS-CoV-2 infection. However, there are many discrepancies that we highlighted in the long discussion. Our purpose was to obtain the most universal prognostic indicators that could be used in clinical practice, regardless of the patient's multiple disease, and we achieved it thanks to a careful analysis of the impact of comorbidities in our models. Furthermore, we believe that our article may contribute to broadening the general knowledge about the influence of COVID-19 on the functioning of the liver.
- “Also, the number of patients included is not very high.”
We agree that our sample is not very high. However, we would like to emphasize that the study was conducted in a main clinical center in Wroclaw, which is one of the largest cities in Poland. Therefore, we allow ourselves to conclude that our group was representative of the Central European region. Moreover, in the calculations, we only considered patients with complete laboratory data on admission to make our results as reliable as possible. Nevertheless, this is a limitation of our study, to which we declare in the dedicated paragraph of the article.
- The manuscript is very difficult to read, as the emphasis of the paper has been put on statistical methods and analyses, and not on the clinical significance of the findings.
Dear Sir/Madame, thank you for this suggestion. A significant part of the article was dedicated to the statistical analysis and the obtained results, but our aim was not to reduce the practical value of our findings. Our purpose was to provide the reader with a reliable explanation on what basis we draw certain conclusions. We believe that due to that approach our study is more objective and the results are easier to revise and repeat by other researchers. It may also be of practical importance to the authors of the meta-analyzes as we clearly described the statistical methods, which can be a main inclusion criterion. Therefore, as we tried to make our article shorter and easier to read, some statistical descriptions have been removed. We hope that in the light of raised points and after shortening the main text You will find our article more informative.
- The English language requires improvements, because in some parts of the manuscript the style impairs the quality of the manuscript, as sentences are difficult to understand.
Dear Sir/Madame, we found your comment adequate. We improved our English, shortened too long sentences. We have tried to make our paper clear and easy to read. We hope that we have met your expectations.
- The manuscript should be rewritten in the manner to be shorter, more informative for the clinicians and easier to understand and follow.
Dear Sir/Madame we considered all suggestions and prepared the paper to make it shorter, more legible and useful in clinical practice. We all grateful for all Your advices. We believe, that they helped us to improve the quality of our article.
Round 2
Reviewer 1 Report
I would like to thank the authors for the revision. The paper has been slightly improved, however some references continue to remain obsolete and too old. They should be changed with newer publications. Additionally the paper continue to remain too extensive and difficult to follow. Section about how clinician can benefit from this research is also needed.
Author Response
We would like to thank the reviewer very much for the constructive comments. Thank you very much. Regarding the concrete suggestions we answered them point-by-point as follows:
1.”I would like to thank the authors for the revision. The paper has been slightly improved, however some references continue to remain obsolete and too old. They should be changed with newer publications.”
Dear Sir/Madame, thank you very much for noticing our efforts to make our work more valuable, notwithstanding that it did not fully meet your expectations. Thank you for your suggestion that some references should be more up to date. We enriched our bibliography with paper from 2021 and 2022. We were particularly focused on the introduction and discussion part, which we refreshed with new concepts on the pathomechanism of liver damage in the course of COVID-19 and updated with the latest knowledge about the frequency of abnormal GGT levels. We hope that considering recent studies made our work more relevant and accurate.
- „Additionally the paper continue to remain too extensive and difficult to follow.”
Dear Sir/Madame, thank you for your comment. We managed to shorten the article, making the results clearer. The discussion part has been updated. We consider it as a short literature review in which we tried to find a context for all of our observations. We were unable to dispose any more content without losing important conclusions. We believe that additionally to the most important observations, as follow predictive values of albumin, AST and TBIL other remarks are valuable, such as the analysis of individual comorbidities in models. They are not the main issue of our survey, but for the readers of the article may be an inspiration for further research. We consider side observations to be a great advantage of our work. Nevertheless, the paper has been shortened and we hope that in this form it will meet your expectations.
- „Section about how clinician can benefit from this research is also needed.”
Dear Sir/Madame, in order to meet your request we added desired information to the conclusion paragraph. We underlined main results that can be useful during the early care of a COVID-19 patient. Thank you for that suggestion, we believe that it allowed us to prepare article more suitable for clinicians.
Round 3
Reviewer 1 Report
The authors have made appropriate changes and I have no further comments
Author Response
Dear Sir/Madame, thank You very much for appreciating our corrections. Thank You very much for all your suggestions at every stage of the review process. We found all of them accurate and valuable. We believe that they contributed significantly to improving the quality of our article.
Considering your recommendation which states that our English requires moderate changes we edited the article focusing on the style and grammatical correctness. We hope that our efforts will be sufficient to meet your expectations.